# Improving Multi-Hop Reasoning in LLMs by Learning from Rich Human Feedback

**Nitish Joshi[1], Koushik Kalyanaraman [2], Zhiting Hu[3], Kumar Chellapilla[2], He He[1], Li Erran Li[2],**

[1]New York University, [2]Amazon Web Services, [3]UC San Diego
nitish@nyu.edu, erranli@gmail.com

## Abstract

Recent large language models (LLMs) have enabled tremendous progress in natural language understanding. However, they are prone to generate confident but nonsensical reasoning chains, a significant obstacle to establishing trust with users. In this work, we aim to incorporate rich human feedback on such incorrect model generated reasoning chains for multi-hop reasoning to improve performance on these tasks. To do so, we collect two such datasets of human feedback in the form of (correction, explanation, error type) for StrategyQA and Sports Understanding datasets[1], and evaluate several algorithms to learn from such feedback. We show that fine-tuning on such small datasets of rich human feedback can improve model's performance of generating the correct final answers, and also improves the model's ability of judging the correctness of it's own answer.

## Introduction

With the onset of large language models (LLMs) (Devlin et al. 2019; Brown et al. 2020), the field has seen tremendous progress on various NLP benchmarks. Among them, the progress has been striking on relatively simpler tasks such as short context or factual question answering (Rajpurkar et al. 2016), compared to harder tasks which require reasoning such as multi-hop question answering (Yang et al. 2018). Even though LLMs may not be best at generating correct reasoning chains or explanations for such hard tasks (Saparov and He 2022), the prompting abilities of LLMs have the potential to provide partially correct (and relevant) facts required to answer the question. Relatedly, recent work has found that without any finetuning LLMs cannot self-correct their reasoning yet (Huang et al. 2023), suggesting the need for human intervention.

Motivated by this, we try to address the following research question — *can we improve reasoning of LLMs by learning from human feedback on model-generated reasoning chains?* Figure 1 provides an overview of our approach — we first prompt the model to generate reasoning chains for multi-hop questions, then collect diverse human feed-

[1]The data and code is available at - https://github.com/joshinh/rich-feedback-reasoning

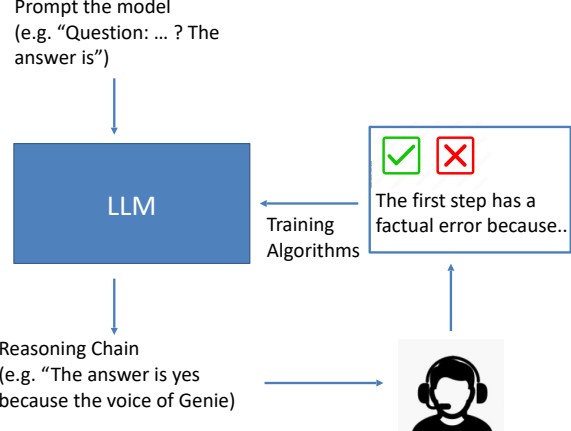

Figure 1: Overview of the process, where we first prompt LLMs to generate reasoning chains for multi-hop questions, collect diverse feedback on the generations including categorical feedback and natural language feedback, and use multiple training algorithms to learn from them.

back on these chains for diagnosis and propose training algorithms to learn from the collected data.

We collected diverse feedback including correction to model's generation, explanation of why the generation was wrong and error type for a total of 2.2k examples from two datasets which we will publicly release. We propose multiple training algorithms to learn from the collected feedback including a multitask algorithm, a variant of self-consistency in chain-of-thought prompting (Wang et al. 2022), and a refinement algorithm where we refine the model generated reasoning chain. We use the proposed algorithms on Llama2 (Touvron et al. 2023) and find that they either improve model's reasoning ability (sports understanding dataset) or perform comparable to in-context learning (strategyQA dataset). More importantly, we find that the fine-tuned model is sometimes better at judging if it's own answer is correct compared to the base (not finetuned) Llama2 model, an important practical ability in order to use LLMs more widely.

Our main contributions can be summarized as: (1) a dataset of rich human feedback including natural language feedback for 2.2k examples for multi-hop reasoning ; (2)

| Question: | Explanation Steps: |
|---|---|
| Q: Has categories of Nobel prizes remained same since Alfred Nobel established them? | **Step 1:** The answer is yes because Alfred Nobel established the Nobel Prizes in 1895.

**Step 2:** The categories of the Nobel Prizes have remained the same since 1895. |

What subquestions need to be answered to answer the original question? This task will help you answer the later tasks. Note that the steps might not be aligned with the subquestions that you write.

Make changes to the provided explanation so that it is correct. You could change spans of text in the given explanations, or add new sentences to make it complete and coherent.

The answer is yes because Alfred Nobel established the Nobel Prizes in 1895. The categories of the Nobel Prizes have remained the same since 1895.

How would you categorize the type of error in the explanation? Use your best judgement to fit the error in the given categories, but if does not match any of the categories please specify other and mention a short description (1-2 sentences) of why the provided explanation is wrong.

For each error type that you check, mention the reason (1-2 sentences) why that error is present. Also include the step number (e.g. step 1) in this description. Check onboarding document for examples.

☐ Factual Error

Reason:

☐ Missing facts

Reason:

Figure 2: The interface used to collect feedback from annotators, displaying all the diverse feedback we collect for an examples from StrategyQA.

novel algorithms to learn from diverse feedback to both improve reasoning performance and which can make LLMs better at judging their own correctness.[2]

## Related Work

**Learning from Feedback.** Learning from human feedback in the form of rewards (Christiano et al. 2017; Ziegler et al. 2019) has become an effective paradigm for improving LLMs (Ouyang et al. 2022; Glaese et al. 2022; Chung et al. 2022b). Most feedback datasets either provide sparse feedback such as binary feedback (Bai et al. 2022a; Ethayarajh, Choi, and Swayamdipta 2022) or provide natural language feedback but for narrow space of tasks such as summarization (Scheurer et al. 2022). In comparison, we create a dataset of rich human feedback including natural language feedback for much harder reasoning tasks.

**LLM self-correction.** In contrast to learning from human feedback, recent works have explored if LLMs can self-correct their answers. Specifically, Madaan et al. (2023) use an iterative feedback and refinement procedure to improve performance; Welleck et al. (2022) introduce self-correctors by separating the generator and the corrector; Bai et al. (2022b) use LLMs to generate feedback based on a 'constitution'. Nevertheless in the context of reasoning, Huang et al. (2023) show that LLMs struggle at self-correction and

---

[2]We believe that the dataset could also be potentially very useful for evaluation in verification (Li et al. 2023) where the task is to identify and describe the error in models' generation.

**Question**: Is the voice of the Genie from Disney's Aladdin still alive?
**Answer**: The answer is no because the Genie was voiced by comedian Robin Williams. Robin Williams died in 2014.

**Question**: Johnny Gaudreau nutmegged the defender. Is this sentence plausible?
**Answer**: The answer is no because Johnny Gaudreau is an American professional ice hockey player. Nutmeg which means passing ball through the opponenet's leg is a term from football.

Table 1: Examples from StrategyQA (top) and Sports Understanding (bottom) used to prompt the language model.

performance might even deteriorate. Given this shortcoming of LLMs' self-correction ability, we collect a rich human feedback dataset for reasoning and demonstrate its utility.

## Data Collection

Here, we describe the details of the feedback we collected and the annotation protocol followed during data collection. We collected feedback for model generations based on two reasoning based datasets: StrategyQA (Geva et al. 2021) and Sports Understanding, part of BigBench (Srivastava et al. 2022). We used GPT-J (Wang and Komat-

| Error Type | StrategyQA | Sports Und. |
|---|---|---|
| None | 17.6% | 31.28% |
| Factual Error | 27.6% | 38.1% |
| Missing Facts | 50.4% | 46.1% |
| Irrelevant Facts | 14.6% | 3.9% |
| Logical Inc. | 11.2% | 5.2% |

Table 2: Percentage of examples in each dataset where the model generation had the particular error type. Note that a example might contain more than one error type.

suzaki 2021) to generate answers for StrategyQA and Flan-T5 (Chung et al. 2022a) to generate answers for sports understanding dataset.[3] In each case, the model was prompted with $k$ in-context examples containing question, answer and explanation such as the ones showed in Table 1, followed by the test question.

Figure 2 shows the interface we used — annotators are given the question, model generated answer and the explanation split into steps. For each question, we collected the following feedback:

**Subquestions**: Decompose the original question into simpler subquestions required to answer the original question. This task was added after a pilot where we found that adding this task helps to 'prime' the annotators and improve quality of the rest of the tasks.

**Correction**: Annotators are provided with a free-form text box pre-filled with the model generated answer and explanation, and asked to edit it to obtain the correct answer and explanation.

**Error Type**: Among the most common types of error we found in the model generations (Factual Error, Missing Facts, Irrelevant Facts and Logical Inconsistency), annotators were asked to pick one or more of the error types which apply to given answer and explanation.

**Error Description**: The annotators were instructed to not only classify the errors but also to give a comprehensive justification for their categorization, including pinpointing the exact step where the mistake occurred and how it applies to the answer and explanation provided.

We used private internal vendors as the annotators. The data collection took place over multiple rounds. We first conducted two small pilots of 30 examples and 200 examples respectively, after which the annotator team were given detailed feedback on the annotation over a video call. We then conducted the data collection over two batches for StrategyQA, and over one batch for Sports Understanding giving periodic feedback throughout — a total of 10 annotators worked on the task over a period of close to one month.

### Dataset Statistics

We gathered feedback on a total of 1565 examples for StrategyQA and 796 examples for Sports Understanding. Table 2 illustrates the percentage of examples that were error-free

---

[3]These models were chosen based on the best state-of-the-art open models at the time of data collection.

**Question** ($q$): Is the voice of the Genie from Disney's Aladdin still alive?
**Model Generation** ($m$): The answer is yes because Genie is voiced by Robin Williams. He is still alive.
**Correction** ($c$): The answer is no because the Genie was voiced by Robin Williams. Robin Williams died in 2014.
**Error Type** ($t$): Factual Error
**Error Description** ($d$): In step 2, the explanation incorrectly mentions that he is still alive when instead he died in 2014.

Table 3: Example of a question, model generation and the feedback we collect for StrategyQA.

in the model generation and the proportion of examples that contained a specific error type. It's worth noting that some examples may have more than one error type.

## Learning Algorithms

For each question $q$, and model generated answer (with explanation) $m$, we have the following feedback collected: correct answer and explanation $c$, type of error present in $m$ (denoted by $t$) and error description $d$, as described in section . Table 3 provides an example with all the feedback.

**Multitask Learning.** A simple baseline to learn from the diverse feedback available, is to treat each of them as a separate task. More concretely, we fine-tune Llama2 with the following objective:

$$\text{maximize} \ \ p(c|q) + p(t|q, m) + p(d|q, m) \quad (1)$$

For each term in Eq 1, we use a separate instruction appropriate for the task (e.g. 'Predict error in the given answer'). We also convert the categorical variable $t$ into a natural language sentence. During inference, we use the instruction for the term $p(c|q)$ ('Predict the correct answer for the given question') to generate the answer for the test question.

**Weighted Self Consistency.** Motivated by the success of self-consistency (Wang et al. 2022) in chain-of-thought prompting, we propose a weighted variant of it. Instead of treating each sampled explanation as correct and considering the aggregate vote, we instead first consider whether the explanation is correct and then aggregate accordingly.

We first fine-tune Llama2 with the same objective as in equation 1. During inference, given a test question $q$, we sample multiple possible answers (with the instruction for $p(c|q)$): $a_1, a_2, .., a_m$. For each sampled answer $a_i$, we use the instruction for the term $p(t|q, m)$ i.e. 'Predict error in the given answer' to identify if it contains error: $t_i = \text{argmax} \ p(t|q, a_i)$. Each answer $a_i$ is assigned a weight of 1 if it is correct, otherwise it is assigned a weight of $\alpha < 1$ (tunable hyperparameter). The final answer is obtained by considering a weighted vote over all the answers $a_1$ to $a_n$.

**Refinement.** In the previous proposed methods, the model directly generates the correct answer $c$ conditioned on the question $q$. Instead, here we propose to refine the model generated answer $m$ to obtain the correct answer for a given

question. More specifically, we first fine-tune Llama2 with the following objective:

$$\text{maximize } p(t; c|q, m) \qquad (2)$$

where ; denotes the concatenation, i.e. error type $t$ followed by the correct answer $c$. One way to view this objective is that the model is first trained to identify the error in given generation $m$, and then remove that error to obtain the correct answer $c$ i.e. first identfy error $t$ in generation $m$ and then refine it to obtain the correct answer $c$.[4]

| Method | StrategyQA | Sports Und. |
|---|---|---|
| In-context learning | $60.40_{1.2}\%$ | $59.66_{4.4}\%$ |
| Multitask Learning | $\mathbf{60.84_{1.0}}\%$ | $74.66_{1.2}\%$ |
| Weighted Self Consistency | $57.92_{1.6}\%$ | $68.16_{0.6}\%$ |
| Iterative refinement | $55.45_{1.8}\%$ | $\mathbf{75.83_{0.5}}\%$ |

Table 4: Performance of learning algorithms on the collected feedback. 'In-context learning' is the baseline using CoT prompting (4-shot) with Llama2.

## Results

For both datasets, we compare all the proposed learning algorithms with the in-context learning baseline using a frozen model (Llama2). All models are evaluated on heldout examples from StrategyQA and Sports Understanding ($\approx$200 ex. from each), where we compute the accuracy of correctly predicting the final answer. For all our experiments, we finetune Llama2 using LoRA(Hu et al. 2021).[5] Note that the model we finetuned (Llama2) is different from the ones which were used for data collection (GPT-J / Flan-T5).

The results are shown in Table 4. As observed, for StrategyQA, the best performing method (multitask learning) performs comparable to in-context learning , but for sports understanding dataset, all proposed methods perform significantly better than the in-context learning baseline. Some examples of generated reasoning chain before and after finetuning can be found in Appendix . We also performed ablations in Table 5 where we removed parts of the feedback during finetuning — we find that removing either error type or error description does not hurt performance significantly (in fact it helps for one dataset) indicating that the collected corrections to reasoning chains were probably more valuable as far as reasoning performance is concerned (compared to error type or error description). Nevertheless, we find that the collected error types provide other benefits such as model being better at judging correctness of its own answer which we demonstrate next.

We investigate how models adapted with human feedback on reasoning mistakes can make them better at identifying errors (Table 6) — this is evaluated by prompting the model

| Method | StrategyQA | Sports Und. |
|---|---|---|
| Multitask | 61.57% | 75.5% |
| Multitask (- error type) | 61.57% | 76.5% |
| Multitask (- error desc.) | 61.13% | 79.5% |

Table 5: Ablation studies — in all cases, the model is still fine-tuned on the corrections.

| Method | StrategyQA | | |
|---|---|---|---|
| | Error | No-Error | Total |
| Majority Label | - | - | 85.7% |
| Base Model | 9.7% | 92.0% | 21.83% |
| Finetuned (ours) | 100% | 29.3% | 89.5% |

Table 6: Accuracy of the model in predicting whether the answer and reasoning are correct. The second row corresponds to Llama2 4-shot CoT prompting, whereas the third row is our multitask finetuned model (avg over 3 seeds).

to predict if its generation contains any error. We prompt the LLM with its own generated answer and reasoning chain (for which we collected feedback) to predict if there is any error. We use the appropriate instruction for the task ('Identify error in the answer')—the same instruction we use when finetuning the models. The model is scored correctly if it predicts 'no error' in the generation if the annotators labeled the example as having no error, or if it predicts any of the error types in the generation (along with 'incorrect' or 'wrong') when the annotators labeled it as having error.[6] Note that we do not evaluate the LLM's ability to correctly identify the error type, but rather if an error is present.

The evaluation is done on a set of 173 additional examples from StrategyQA for which we collected feedback, which are not seen during finetuning. 4 examples out of these are reserved for prompting the language model (second row in Table 6).[7] We compute accuracy separately for examples with and without an error in the generation (denoted by 'Error' and 'No-Error' respectively). We observe that our finetuned models are better at judging the correctness of generating reasoning chains both compared to the base model (in-context learning) and a majority label baseline. [8]

## Conclusion

We curate human feedback datasets with fine-grained error corrections—both categorical (error types) and free form (error description)—an alternative way to improve the reasoning abilities of LLMs. Experimental results corroborate that human feedback on reasoning errors can improve performance and calibration on challenging multi-hop questions even with a small amount of feedback.

---

[4]A potential extension of this is to use it iteratively till the model predicts there is no error in the generation i.e. use $c$ as $m$ for the next iteration, and the (i-1)th answer $c_{i-1}$ is correct if $t_i =$ no error.

[5]More experimental details can be found in the Appendix.

[6]In most cases, we find that LLMs do not deviate from these options since they were finetuned to produce one of these outputs.

[7]Exact prompts can be found in the Appendix.

[8]Note that the 'total' accuracy is better for the finetuned model since a lot more generated outputs have an error in them.

## Acknowledgement

We thank Hanlin Zhang for the helpful discussion, data collection, and experimentation.

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

## Experimental Details

For all the experiments, we use a batch size of 16, learning rate $1e-4$ with Adam Optimizer and a total of 5 epochs of training. To finetune large models, we use a parameter efficient finetuning method — LoRA (Hu et al. 2021). We apply LoRA to the projection matrices for the queries and values. We set the rank $r = 16$, $\alpha = 16$ and the dropout to 0.05 as the LoRA hyperparameters. For the main experiments in Table 4, we report mean and standard deviation over 3 runs. All models were trained on a A100 GPU.

## Prompts

For the experiment to judge the correctness of the generated reasoning chain, we use a 4-shot prompt for the baseline which is shown in Table 7.

Q: Can preventive healthcare reduce STI transmission? The answer is yes because STIs are caused by bacteria and viruses. Preventive healthcare can reduce the risk of contracting STIs.
E: The output is correct.
Q: Is cow methane safer for environment than cars? The answer is no because Cars produce carbon dioxide and methane. Methane is a greenhouse gas.
E: The output is wrong.
Q: Does the central processing unit usually have a dedicated fan? The answer is yes because The central processing unit (CPU) is the main component of a computer. The CPU is the part of the computer that processes information. The CPU is usually cooled by a fan.
E: The output is correct.
Q: Did Al Unser Jr. win the Space Race? The answer is no because Al Unser Jr. was the first American to win the Indianapolis 500. The Space Race was a competition between the United States and the Soviet Union to be the first to put a man on the moon.
E: The output is wrong.
Q: Is the United States Capitol located near the White House?

Table 7: The 4-shot prompt used for the baseline (first row) in Table 6.

## Example Generations

In this section, we illustrate some examples of the generated reasoning chains before and after finetuning.

**Before finetuning**:
Q: Jacob Markstrom backhanded a shot. Is this sentence plausible?
A: The answer is no because Jacob Markstrom is a Swedish professional ice hockey player. In ice hockey, a backhanded shot is a shot that is struck with the back of the stick.
**After finetuning**:
Q: Jacob Markstrom backhanded a shot. Is this sentence plausible?

A: The answer is yes because Jacob Markstrom is a professional ice hockey goaltender. Backhanding a shot is a common technique in ice hockey.

In the above example, the base model generated a correct reasoning chain but had a logical inconsistency and still predicted 'no'. After finetuning, the model predicts 'yes' which is the correct answer.

**Before finetuning**:
Q: Brandon Lowe took a left footed shot. Is this sentence plausible?
A: The answer is yes because Brandon Lowe is an American professional baseball player. In baseball, a left footed shot refers to a batted ball that is hit with the left foot.
**After finetuning**:
Q: Brandon Lowe took a left footed shot. Is this sentence plausible?
A: The answer is no because Brandon Lowe is a baseball player. Baseball players do not take left-footed shots.

In the above example, the base model has a factual error where it hallucinates a left footed shot in baseball. After finetuning, the model predicts the correct reasoning chain and the correct answer 'no'.

**Before finetuning**:
Q: Evgeni Malkin passed the puck. Is this sentence plausible?
A: The answer is no because Evgeni Malkin is a Russian ice hockey player. In ice hockey, a puck is a small, hard rubber disc that is used to play the game.
**After finetuning**:
Q: Evgeni Malkin passed the puck. Is this sentence plausible? A: The answer is yes because Evgeni Malkin is a professional ice hockey player. Passing the puck is a common action in ice hockey.

In this example, similar to the first one, there is a logical inconsistency in the base model — the model predicts 'no' even though the (correct) predicted reasoning chain would suggest 'yes'. After finetuning, this error is fixed.

## Limitations

In this work, we only consider four types of error types (Factual Error, Missing Facts, Logical Inconsistency, Irrelevant Fact) and leave a more flexible feedback categorization for future work. Additionally, we also collect feedback for only a small amount of examples ($\approx$1.5k for StrategyQA and $<$ 1k for Sports Understanding).