# OpenReview forum: "Improving Multi-Hop Reasoning in LLMs by Learning from Rich Human Feedback"
_AAAI.org/2024/Workshop/NuCLeaR — NuCLeaR 2024_

### Official Review · Reviewer_uB84 · 2023-12-04
**A comprehensive work that tackles difficult questions**

**Rating:** 9
**Confidence:** 5

**Review:**

## Overall clarity
### Pros
Clear language expression. Well-placed references and citations. Pertinent literature review and related work. A handy appendix that further details the data curation and evaluation steps.

### Cons
Minor grammatical and punctuation mistakes.

## Overall originality
### Pros
The work is original on two fronts: it proposes two new datasets and a new learning algorithm for better handling reasoning in large language models (LLMs).

### Cons
The work could have benefitted from an architectural diagram of the fine-tuning and innovation applied at the learning algorithmic level.

## Overall quality
### Pros
Clear and detailed evaluation results and ablation studies. The dataset collection process is explained in depth as well. Result analysis of the performance and the novelty of the technique with respect to LLMs judging their generations is also highlighted.

### Cons
More a bonus than a critique: it would greatly benefit the work to be able to shed light on a comparable performance with other famous LLM generations besides LLaMA (e.g., ChatGPT, Falcon, etc.)

## Overall conclusion
An excellent contribution in reasoning for large language models.

---

### Official Review · Reviewer_jf2u · 2023-12-06
**Good annotated datasets, weak in evaluation**

**Rating:** 6
**Confidence:** 5

**Review:**

This paper tackles an important research question: how to improve LLM's reasoning and multi-hop QA abilities. Its main contribution is a collection of newly annotated human-feedback data for over 2k examples about correctness, error types and explanations in two tasks: StrategyQA and Sports understanding. However, the algorithms proposed are not particularly novel and the evaluation is especially weak.

Specifically, there are some major weakness on the evaluation, which raises more questions on the approach/algorithm used:

1) The performance gains were inconsistent across the two datasets
2) When evaluating the prediction of the final correct answers, the main evaluation metric is the accuracy. Since the majority of examples in the test sets have errors in the reasoning chains (see Table 2 in paper), this is a highly imbalanced classification problem. The use of accuracy alone is misleading since the models can be biased towards majority "incorrect" predictions. Other metrics such as Precision and Recall or F1 score could be used and reported to get a better error breakdown picture of the model performance.
3) There is no human or expert evaluation to deeply assess reasoning improvement qualitatively. This could shed more lights on the actual improvement
4) The proposed approaches are basic. Meta-learning tailored for reasoning could be more effective.
5) No ablation study is done to remove different feedback signals (e.g, error type vs. error description vs. corrections) from training. Such experiments could reveal which forms of feedback are most valuable for enhancing reasoning performance.

---

### Official Review · Reviewer_1DLv · 2023-12-07
**The paper studies how to improve  LLMs using human feedback for  multi-hop reasoning tasks. The paper is well written but lacks some necessary aspects.**

**Rating:** 6
**Confidence:** 4

**Review:**

**Summary:** The goal of the work is to enhance the performance of LLMs in tasks requiring multi-hop reasoning through human feedback.

**Strengths:**
- Introduces a dataset of human-feedback
- Presents methods and objectives to learn from the feedback dataset

**Weaknesses:**
- The applicability of the method seems to be task-specific. It is not clear how methods will be performed in unseen datasets.
- The collection of human preferences is costly; whereas AI feedback is cheaper. The paper does not address this issue.
- The proposed objectives are variants of existing techniques.
- RL-based objectives are not compared.
- The error-types look limited. There can be other types of errors.

---

### Decision · Program_Chairs · 2023-12-11

Accept